# Genetic Algorithm for Curriculum Design in Multi-Agent Reinforcement Learning

**Yeeho Song**[*]
School of Computer Science
Carnegie Mellon University
United States of America
yeehos@andrew.cmu.edu

**Jeff Schneider**
School Of Computer Science
Carnegie Mellon University
United States of America
jeff4@andrew.cmu.edu

**Abstract:**
As the deployment of autonomous agents in real-world scenarios grows, so does the interest in their application to competitive environments with other robots. Self-play in Reinforcement Learning (RL) enables agents to develop competitive strategies. However, the complexity arising from multi-agent interactions and the tendency for RL agents to disrupt competitors' training introduce instability and a risk of overfitting. While traditional methods depend on costly Nash equilibrium approximations or random exploration for training scenario optimization, this can be inefficient in large search spaces often prevalent in multi-agent problems. However, related works in single-agent setups show that genetic algorithms perform better in large scenario spaces. Therefore, we propose using genetic algorithms to adaptively adjust environment parameters and opponent policies in a multi-agent context to find and synthesize coherent scenarios efficiently. We also introduce GenOpt Agent—a genetically optimized, open-loop agent executing scheduled actions. The open-loop aspect of GenOpt prevents RL agents from winning through adversarial perturbations, thereby fostering generalizable strategies. Also, GenOpt is genetically optimized without expert supervision, negating the need for expensive expert supervision to have meaningful opponents at the start of training. Our empirical studies indicate that this method surpasses several established baselines in two-player competitive settings with continuous action spaces, validating its effectiveness and stability in training.

**Keywords:** Reinforcement Learning, Multiagent Learning, Curricular Learning

## 1 Introduction

Competitive multi-agent reinforcement learning (RL) has gained interest for its potential in various fields, such as gaming, robotics, finance, and cybersecurity, where agents need to outperform others. To discover new competitive strategies, self-play is often used to explore the environment with RL agents. Most self-play algorithms train ego agents against a population of opponent policies to avoid overfitting to a specific opponent. [1, 2, 3].

When training against a population of policies, selecting the policies most useful for training is challenging. Some works use game-theoretic methods to select opponents [4, 5], but finding approximate Nash Equilibria in setups with large search spaces is costly. Others use curriculum learning to select opponent policies [6], but relying on random sampling to find candidates is not scalable with large search spaces. Therefore, we propose using genetic algorithms, which have proven efficient for searching and generating scenarios with large scenario spaces in single-agent domains [7].

Another issue with self-play is training instability due to the adversarial vulnerability of RL policies [8, 9, 10]. RL agents often overfit to a specific opponent's weakness, resulting in non-generalizable

---

[*]yeehos@andrew.cmu.edu, Robotics Institute

8th Conference on Robot Learning (CoRL 2024), Munich, Germany.

behaviors. For example, when training against a mujoco-ant agent trained to play sumo [11], agents would learn to win by waving their legs to confuse the opponent, instead of the non-trivial solution of pushing the opponent. Exploitation destabilizes self-play, especially during the initial phases when the opponent's policy is underdeveloped. Initiating opponents with expert demonstrations [3] to avoid this issue can be costly. We, therefore, introduce GenOpt Agents, which are environment-agnostic open-loop agents optimized by genetic algorithms to match the performance of the ego agent.

To summarize, our approach utilizes genetic algorithms to efficiently search and generate an optimal training curriculum in expansive search spaces, while GenOpt Agents improve training stability. An ablation study provides insights into our methodology's effectiveness and design choices. Figure 1 summarizes our approach. Supplementary information and codes can be found in `https://github.com/yeehos/GEnetic-Multiagent-Selfplay.git`

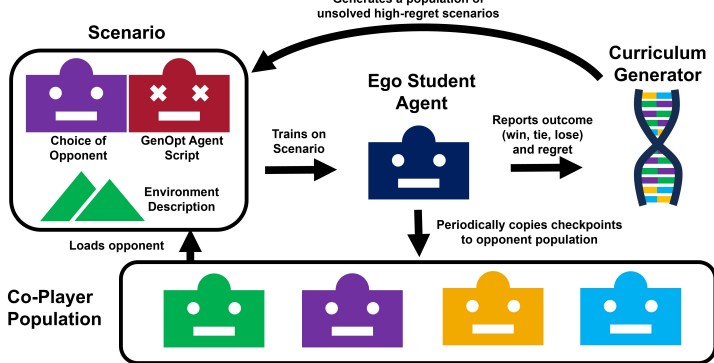

Figure 1: Overview of our proposed approach: During training, the ego student agent is trained against a scenario describing the choice of opponent, action script for the GenOpt Agent, and the choice of environment parameters. Our curriculum records regret and win/lose/tie outcomes for each scenario. Based on the performance of the scenarios in the previous population, our curriculum generator uses a genetic algorithm to generate a new population of scenarios to be used to train the agent for an epoch at each fixed interval of steps.

## 2 Related Works

**Game Theoretic Approach to Self-Play**

Training an RL agent in competitive multi-agent environments requires an opponent. Unlike rule-based opponents [12] or expert demonstrations [13], self-play [14, 15] agents to autonomously explore and discover competitive behaviors. However, training against only one RL opponent can suffer from the instability of opponent behavior changing over time and ego agent overfitting to a specific policy [16]. Therefore, Fictitious Self-Play (FSP) [17, 18, 1] proposes to train against a population of opponents. In deep RL, this opponent library is often compiled by saving checkpoints of the ego policy during training [3, 19].

Expanding on these findings, game theoretic approaches like Policy Space Response Oracle (PSRO) [4, 3, 19] utilize Nash equilibrium [5], a stable point in a multi-agent game where no player can unilaterally change their strategy to improve their payoff. PSRO calculates an approximate Nash strategy to determine the mixture of checkpoints to load as opponents. However, computing an approximate Nash strategy can be computationally intensive if the policy space is large. Moreover, effectively learning policies in complex, non-stationary, and partially observable multi-agent environments remains a significant challenge. While some approaches use ensemble learning to split the learning task [20] [21], they are limited to simpler, discrete action setups. This paper approaches this issue with a curricular viewpoint.

**Curricular Reinforcement Learning**

When it comes to learning a policy for a difficult problem, Curricular RL suggests that agents can learn faster by first being exposed to simple tasks and scenarios and gradually progressing to similar but slightly more difficult tasks. [22, 23, 24, 25, 26, 27, 28, 29, 30, 31].

Various papers explore curriculum optimization in the single-agent domain, such as Bayesian Optimization [32], teacher agents [33], and genetic algorithms [34, 35, 36]. Genetic operations make scenarios similar to each other, thus achieving curriculum learning by facilitating the transfer of skills. Many genetic approaches rely on mutations, which make small alterations to the encoding of a scenario, in order to populate a curriculum with similar scenarios.

On the contrary, the Genetic Curriculum (GC) [7] uses population-wide genetic operations, such as crossover, for curricular RL. Crossover involves merging sequences encoding across a population, helping the transfer of skills within the curriculum by enhancing similarity among the scenarios. While successful in generating an effective curriculum in a large scenario space, GC requires costly evaluation steps to validate scenarios. Additionally, GC cannot regulate scenario difficulty.

One way to regulate difficulty is to use regret, which quantifies the gap between the current obtained return with the maximum possible return on the scenario. Showing success in various single-agent domains [37, 38], regret has been applied to curriculum learning in a multi-agent setup by Multi-Agent Environment Design Strategist for Open-Ended Learning (MAESTRO) [6]. While MAESTRO introduced optimization of both environmental parameters and opponent selection to guarantee robustness, it is limited by relying on domain randomization to uncover new scenarios. Studies in GC [7] suggest this can lead to suboptimal performance. Unlike those generated by population-wide genetic operations, scenarios generated by domain randomization may not be similar to each other, making transfer of skills difficult and learning slow. We, therefore, combine both population-wide genetic operations with regret regulation for multi-agent curriculum generation.

**Open-Loop Opponents for Multi-Agent Self-Play**

Self-play can be unstable, especially at the beginning of training when the opponent is not well-trained to make effective moves in the game. While some approaches involve hand-crafted opponents [3] or agents trained via imitation learning from expert data [39], such supervision can be costly.

No-OP (No Operation) agents [40], which do not take any actions, can reduce the need for expert supervision. However, their simple and limited behavior limits the approach to certain types of environments. For instance, a walking robot will fall if there is no torque in the joints, while a plane with no control input will eventually crash by losing speed and altitude due to drag. Our GenOpt addresses such issues by introducing agents that take optimized actions.

## 3 Approach

### 3.1 Preliminaries

An RL problem setup is typically represented as a tuple in a Markov decision process: $[S, A, P, r, \gamma]$, where $S$ is the state space of a problem, $A$ is the action space, $P$ is the transition dynamics, $r$ is the return of a state-action pair, and $\gamma \in [0, 1)$ is the temporal discount factor. The agent's policy, $\pi(a \mid s)$, maps states $s \in S$ to actions $a \in A$. The utility of a policy $\pi$ is the expected return is denoted as $J(\pi) = \mathbb{E} \sum t \gamma^t r(s_t, \pi(s_t))$. During training, an RL algorithm optimizes the policy with respect to the data it has collected.

In our multi-agent setup, we consider that the utility of a policy is dependent on both the opponent's policy $\pi_{opp}$ and the environment $\psi$, denoted as $J(\pi_{ego}, \pi_{opp}, \psi)$. At each epoch consisting of a fixed number of time steps, we save the current version of our ego agent and add it to the library of possible opponents to choose from $\Pi_{opp}$. We describe the environment with parameters, $\psi$. We define a scenario $\xi$ as an opponent policy - environment pair: $\xi = \{\pi_{opp}, \psi\}$. The goal of our curriculum generator at a given timestep is to generate a population of scenarios, $\Xi_{train} = \{\xi_0, \xi_1, ..., \xi_n\}$, where optimizing the ego policy $\pi_{ego}$ with respect to the curriculum $\Xi_{train}$ will result in a policy with the behaviors we desire.

$$\pi^* = \max_\pi J(\pi_{ego}, \Xi_{train})$$

### 3.2 Problem Formulation

For the multi-agent setup, we want our algorithm to continuously explore the game by visiting scenarios that it does not do well to learn new competitive behaviors. We design our curriculum generator around this idea by first formulating multi-agent learning as a zero-sum game $G(\pi_{ego}, \xi)$ between

the $\pi_{ego}$ and the curriculum generator. $G(\pi_{ego}, \xi) = 1$ if $\pi_{ego}$ wins the scenario $\xi$, $G(\pi_{ego}, \xi) = 0$ if it was a tie, and $G(\pi_{ego}, \xi) = -1$ if loses. The opponent $\pi_{opp}$ and the environment parameters $\psi$ are determined by the curriculum's scenario $\xi$. The solution for a finite zero-sum two player game is minimax;

$$G^* = \min_{\Xi} \max_{\pi_{ego}} G(\pi, \Xi)$$

Therefore, our curriculum generator will generate a population $\Xi$ that minimizes $G(\pi_{ego}, \Xi)$

### 3.3 Genetic Algorithm for Curriculum Generation

In this section and onward, we explain how our proposed algorithm, GEnetic Multi-agent Self-play (GEMS), as described in Algorithm 1, generates curriculum.

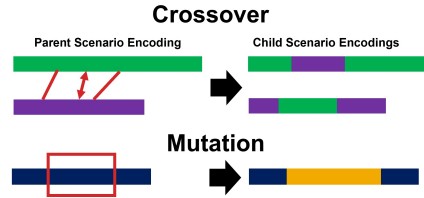

Figure 2: Visualization of crossover and mutation. Crossover replaces encoded segments between two parent scenarios. Mutation changes an encoded segment of a parent with a random sequence.

To generate a curriculum, GEMS uses a genetic algorithm due to its proven efficiency in large scenario spaces and its flexibility in scenario encoding. At the beginning of the training, as in Line 5 of Algorithm 1, GEMS randomly initializes curriculum $\Xi_{train}$. $\pi_{ego}$ trains on the $\Xi_{train}$ for an epoch consisting of a fixed number of steps. For each scenario, our algorithm records whether the agent has won, lost, or tied, along with the regret estimated by positive value loss [37, 41].

At the end of each epoch, GEMS harvests scenarios from the current curriculum to generate the next curriculum as in line 14 of Algorithm 1. The fitness function, $p(\xi)$, defines the probability of a scenario being harvested. Since the GEMS is trying to minimize $G(\pi_{ego}, \Xi)$ of the generated curriculum, fitness is proportional to $(1 - G(\pi_{ego}, \xi))$. At the same time, GEMS needs to generate scenarios with high information potential, measured as regret, $\delta(\xi)$. Therefore, fitness is set as $p(\xi) \propto \delta(\xi)(1 - G(\pi_{ego}, \xi))$.

From these harvested scenarios, GEMS uses crossover and mutation (see Figure 2) to create an offspring population consisting of sequences of $\xi$ similar to the ones harvested. Crossover occurs when the algorithm takes a random segment from one parent scenario encoding and swaps it with a random segment from another parent scenario encoding. The mutation is when the algorithm selects a random segment from a parent scenario encoding and swaps it with a segment from a randomly generated scenario encoding. Scenarios $\xi$s generated by the genetic algorithm will inherently be similar, aiding in the transfer of skills and, consequently, a faster rate of convergence when used as a curriculum. Detailed operations for crossover and mutation are shown in Appendix E.

Unlike GC, which has separate evaluation steps, GEMS calculates fitness from the performance measured during training, reducing the computational cost. In addition, GEMS regularizes difficulty.

### 3.4 GenOpt Agent and Scenario Space

Self-play in deep RL often suffers from training instability due to ego agents exploiting the adversarial vulnerability of the opponents. While using open-loop agents can mitigate this issue, previous approaches used expert supervision to design open-loop behaviors, which can be costly [40]. To allow an open-loop agent that continuously evolves to match the ego agent's performance during training without expert supervision, we introduce GenOpt Agent, denoted as $\pi_{\varnothing}$, that optimizes open-loop behaviors using genetic algorithms without supervision.

$\pi_{\varnothing} = \{(t_0, a_0), (t_1, a_1), (t_2, a_2).....\}$ encodes the opponent as a list of non-fixed lengths describing which actions to take in an open-loop fashion. For example, at timestep $t_1 < t < t_2$, the opponent takes action $a_1$. As genetic algorithms can be agnostic to the length of encoding, the encoding length does not need to be pre-specified and is optimized by the curriculum generator without expert supervision. GenOpt's open-loop nature makes it less vulnerable to opponent-induced confusion, but unlike No-OP, GenOpt is competitively optimized to match the ego agent's performance, enhancing its applicability across diverse environments.

Using the features above, GEMS encodes a scenario as $\xi = \{i_\pi, \pi_\varnothing, \psi\}$ $i_\pi$ is an integer representing the i-th checkpoint to load as the opponent $\pi_{opp}$. If $i_\pi = 0$, it indicates that no agent will be loaded, and instead, a GenOpt Agent $\pi_\varnothing$ will be used as the opponent. $\psi$ denotes which environment parameters to load. It should be noted that whether it is used or not, the encoding for the GenOpt Agent $\pi_\varnothing$ is always present in a scenario $\xi$. This ensures that the optimization and the memory about a GenOpt Agent $\pi_\varnothing$ are not lost during the genetic operations and can always be brought back if needed throughout the training.

## 4 Experiments

### 4.1 Benchmarks

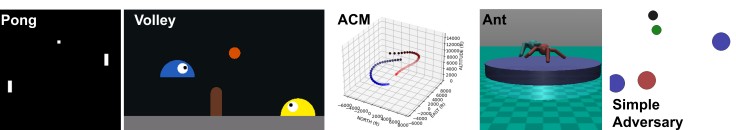

Figure 3: Screenshots of Pong, Volley, ACM (dots marking trajectories), Ant, and SimpleAdversary

We evaluate our algorithm in Pong, Volley, ACM, Ant, and SimpleAdversay to assess its performance in different levels of complexity. More environment details are in Appendix B.

***Pong*** is a 2-player, continuous-action space version similar to Atari Pong [42]. Each player controls a paddle to hit a ball back and forth, aiming to score by getting the ball past the opponent. The scenario set the players' sides and the ball's initial velocity. Simple 1-D movement and fixed horizontal ball speed focus make dynamics easy to learn and relatively emphasize the game-theoretic aspect of the solution more than other benchmarks.

***Volley*** is based on [43]. Each player uses continuous action input to control an avatar,

**Algorithm 1** GEnetic Multi-agent Self-play

1: **Initialize** Policy $\pi_{ego}$
2: #*Select size of curriculum, $M_{train}$*
3: #*Select mutation rate, $p_\mu$*
4: **Input** $M_{train}, p_\mu$
5: **Initialize** Curriculum $\Xi_{train}$
6: **while** True **do** outcome = [] regrets = []
7:    #*Train $\pi$ with $\Xi_{train}$ by exploring scenario $\xi$*
8:    **for** $\xi$ **in** $\Xi_{train}$ **do**
9:       $G(\pi, \xi), \delta$ = Train($\pi_{ego}, \xi$)
10:       outcome.append($G(\pi_{ego}, \xi)$)
11:       regrets.append($\delta$)
12:    **end for**
13:    #*Harvest Examples*
14:    $\Xi_{seed}$, utility = **harvest**($\Xi_{train}$,outcome,regret)
15:    #*Generate New Curriculum*
16:    $\Xi_{train}$ = **crossover**($\Xi_{seed}, M_{train}$,utility)
17:    $\Xi_{train}$ = **mutation**($\Xi_{train}, p_\mu$)
18:    **save**($\pi_{ego}$) #*Regularly save checkpoint*
19: **end while**

moving left, right, and jumping. The goal is to bounce the ball to land on the other side of the map across the net. The scenario specifies the players' sides and the initial ball velocity. Volley incorporates gravity, elastic collisions, and a net, increasing complexity and skill requirements.

***ACM*** is a simulated dogfighting environment. Each player flies an airplane in 3D space, aiming to position its nose toward the opponent without crashing to the ground. The scenario describes the spawning airplanes' positions, postures, and velocities. With physics simulated by JSBSim [44], a high-fidelity simulator widely adopted in autonomous aircraft and aircraft controls research due to its accurate aerodynamic modeling [45], and direct control over the aircraft operating in 3D space with no stability assist, ACM is a complex environment that needs well-trained skills to master the game with non-linear dynamics. Additionally, since the game's objective is to maneuver and position the ego plane in relation to the opponent, the scenario, encoding spawn locations and orientations of the planes, significantly influences the game's outcome. ACM tests whether algorithms are robust when environmental factors outside their control give an unfair advantage to one of the agents.

***Ant*** is a simulated Mujoco environment based on OpenAI's Competitive Gym environment. [11]. The game's goal is to flip the opponent or push the opponent out of the ring. Ant challenges the agent to learn multiple different skills to master the game. For example, while ramming is often the only way to win an opponent that stands still, those who ram are vulnerable to those who flip the opponent. However, flipping is not helpful against opponents that stand still, as an opponent keeping all its legs on the ground leaves no room for flipping. The agents must learn various skills and know how to mix and use them to win a game. To the best of our knowledge, we are the first paper to solve the Ant problem by training from scratch without expert supervision or pretrained weights.

*SimpleAdversary* is a Multi Particle Environment from [46]. As in the Figure 2, the environment features one adversary (red), two good agents (blue), and two landmarks (green and black). Agents are rewarded based on their proximity to the target landmark, with opposing goals for good agents and the adversary. Only good agents know the target's location. SimpleAdversary tests offers asymmetrical environment with goals, action and observation space being different per role, along with added challenge of collaborative-competitive settings with multiple agents.

## 4.2 Baseline Algorithms

We use PSRO as one of the baselines for comparison. While many algorithms have emerged from PSRO, such as Rectified PSRO [47], Pipeline PSRO [48], or Anytime PSRO [49], these are fundamentally based on PSRO's framework. They share the underlying structure of generating Nash strategy opponents without considering the ego agent's performance to present the tasks in a gradual, easy-to-learn fashion. Therefore, we utilize PSRO as a characteristic example to highlight the limitation of this assumption. While PSRO necessitates additional steps to evaluate each policy to run the meta-solver during training, we report performance without considering the extra evaluation steps that PSRO requires. We believe that presenting PSRO's performance without factoring in these additional steps provides a comprehensive comparison, not only against PSRO itself but also against newer subsequent algorithms such as XDO [50] and NAC [51], which aim to reduce the computational cost associated with policy evaluation.

We include GC [7] as a comparison against approaches that use a genetic algorithm to generate a curriculum. We include SPDL [26] to compare our approach against algorithms controlling environment parameters and actively regulating difficulty levels during curriculum generation. For fairness in the multi-agent domain, GC and SPDL are running with Fictitious Self-Play (FSP) [1], labeled GC+FSP and SPDL+FSP, where the curriculum generator can choose opponents from saved checkpoints. While other single-agent curricula RL regulate difficulty levels, such as using regret, we decide to use an approach more relevant to the multi-agent domain by including MAESTRO [6]. MAESTRO represents a state-of-the-art approach to optimizing environment-opponent choices in the multi-agent domain using regret to regulate the difficulty level of the scenarios. Finally, we include FSP as a baseline comparison for simple population-based multi-agent RL. FSP loads environmental parameters by domain randomization.

For all algorithms, we use a publicly available implementation of Soft Actor Critic (SAC) [52, 53] and Proximal Policy Optimization (PPO) [54, 55] as the base strategy explorers.

## 4.3 Evaluation and Hyperparameters

We conduct experiments to evaluate each method's effectiveness using ten random seeds for Pong, Volley, and 5 for the more complex ACM and Ant benchmarks. The training duration varied from 5 to 15 days, with computationally intensive algorithms like ACM requiring the most time. Given the impracticality of exploitability analysis in complex environments, performance was assessed through competitions against five baselines and our GEMS across 200 games with randomized environment parameters, utilizing Round Robin format for both interim (every 2.5e5 timesteps for SimpleAdversary, every 5e5 timesteps for the rest) and final evaluations. Ablation studies were performed with five seeds to validate our design decisions further, comparing ablated models against fully trained baselines.

While PSRO, GC, and SPDL involve additional simulation steps for curriculum generation, we report results based on the exploration steps taken by each agent for easier comparison. We train each algorithm for 7e6 steps in each benchmark, and 3.5e6 steps for the SimpleAdversary. See Appendix B for implementation details and hyperparameter tuning results.

## 5 Results

### 5.1 Round Robin Results

As summarized by the round robin results in Table 1, Our approach outperforms all baselines in the win and lose rates across benchmarks. See Appendix A for comprehensive results.

|  | Pong | Volley | ACM | Ant | SimpleAdversary |
|---|---|---|---|---|---|
| **FSP** | 59 : 1 : 39 | 40 : 5 : 55 | 37 : 49 : 14 | 38 : 28 : 34 | 50 : 0 : 50 |
| **PSRO** | 56 : 1 : 42 | 40 : 5 : 55 | 35 : 51 : 14 | 37 : 21 : 42 | **53** : 0 : **47** |
| **GC+FSP** | 58 : 1 : 42 | 46 : 7 : 47 | 23 : 34 : 43 | 36 : 24 : 39 | 49 : 0 : 51 |
| **SPDL+FSP** | 31 : 0 : 69 | **47** : 24 : **28** | 28 : 43 : 29 | 37 : 21 : 42 | 46 : 0 : 54 |
| **MAESTRO** | 21 : 0 : 79 | 42 : 6 : 52 | 10 : 23 : 67 | 34 : 22 : 43 | 49 : 0 : 51 |
| **GEMS** | **72** : 3 : **25** | **48** : 24 : **28** | **41** : 52 : **8** | **44** : 36 : **21** | **53** : 0 : **47** |

Table 1: Mean Win:Tie:Lose Ratio (%) of algorithms against baseline algorithms and ours *(GEMS)*. In bold are the statistically significant highest win rates and lowest lose rates.

While MAESTRO performs comparable to FSP in Volley, it does not perform well in other benchmarks. GC [7] suggested that relying on random exploration instead of curriculum generation can be limiting. We will confirm that this limitation extends to the multi-agent setup in our ablation study.

While PSRO performs well in Pong, where dynamics are simple and approaching game-theoretical solutions is relatively more important, PSRO's effectiveness diminishes in more complex environments like Volley, ACM, and Ant. This highlights the significance of a curriculum that aids in learning new skills against a broad spectrum of opponents. Also, while PSRO's game theoretic approach allows it to outperform most of the baselines in an asymmetrical setup such as SimpleAdvesary, as shown in Table 4, our GEMS outmatches PSRO via flexible architecture applicable to such setups as well.

Despite both being curricular methods, GC+FSP and SPDL+FSP exhibit different performances. SPDL+FSP can regularize scenario difficulty and outperforms GC+FSP in more complex environments such as Volley, ACM, and Ant compared to PONG. GC+FSP, on the other hand, has a robustness guarantee unlike SPDL+FSP, and outperforms SPDL+FSP in simple benchmarks such as Pong, where difficulty regularization is not important. However, both suffer from ACM benchmark as neither are designed for multi-agent setups.

MAESTRO's performance is comparable to FSP in Volley but is less effective in other benchmarks, especially in ACM and Ant. This highlights the limitations of relying on random exploration in expansive scenario spaces. Our ablation study further supports this finding in the multi-agent context.

## 5.2 Evolution of Generated Curriculum

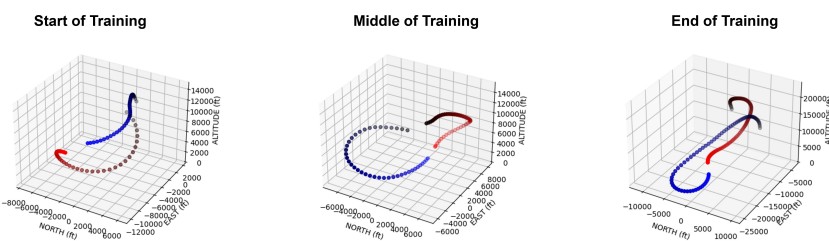

Figure 4: Evolution of scenarios generated by our GEMS. Each dot marks the position of red and blue aircraft at 1-second intervals. The color of the markers starts from black to blue for student aircraft and black to red for the opponent aircraft.

To demonstrate the curriculum evolution, Figure 4 shows characteristic scenarios generated at 1e6, 3e6, and 7e6 steps with the ACM benchmark. Initially, to accommodate the untrained ego agent, our algorithm generates GenOpt that spawns close and circles around the ego agents, offering non-trivial and easy opponents for the ego agent to practice basic tracking. As the training progresses, our GEMS starts to generate scenarios with interesting learning examples, such as the opponent chasing the ego agent for a head-on pass. This gradual complexity increase enables the agent to tackle complex situations by training's end, such as a much longer chase shown in the figure. For extensive visualizations and comparisons, see Figure 7 in the appendix.

## 5.3 Ablation Study

We conduct an ablation study on our GEMS to demonstrate the empirical effects of our design choices. **NoRegret** experiment excludes the regret term ($\delta$) when calculating the fitness function to observe how effectively regret regulates the difficulty level of scenarios during training. **NoGenetic** experiment employs the approach of MAESTRO [6] instead of using a genetic algorithm to generate a scenario population. Scenarios are individually selected from a replay buffer, and new scenarios are added through a random search, deviating from the batch generation of a genetic algorithm. **NoCrossover** experiment disables the crossover function of the genetic algorithm, relying solely on mutation for scenario search. **NoGenOpt** experiment disables the option to use a GenOpt Agent during training. **NoVic** experiment disables the fitness function from utilizing $G(\pi_{ego}, \psi)$, focusing solely on maximizing regret. Lastly, **No-OP** replaces GenOpt Agents with No-OP agents, which takes no actions.

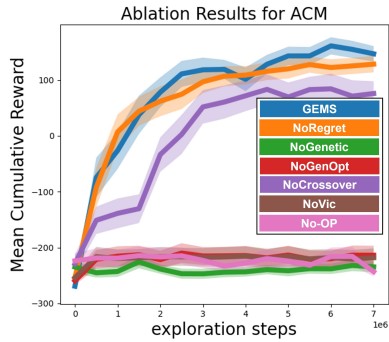

Figure 5: Ablation study training curve

While NoRegret initially mirrors GEMS in training performance, it plateaus earlier. This marks that while regret becomes crucial for adjusting curriculum difficulty towards equilibrium, its impact on training efficacy is less significant compared to GEMS's other design features.

One key aspect of GEMS is the use of genetic operations. GC [7] showed that genetic algorithms outperform random exploration in single-agent settings by efficiently identifying challenging scenarios similar to each other, facilitating effective task generation. The ablation study shows that this advantage also applies to the multi-agent domain, where even solely relying on mutation (NoCrossover) is more effective than random exploration (NoGenetic).

| | |
|---|---|
| **MAESTRO** | -226.845±25.332 |
| **GEMS** | **139.080±32.768** |
| **NoRegret** | 128.416±14.724 |
| **NoGenetic** | -234.64±9.595 |
| **NoCrossover** | 75.808±21.963 |
| **NoVic** | -218.267±16.605 |
| **NoGenOpt** | -213.751±16.907 |
| **No-OP** | -243.172±8.521 |

Table 2: Mean return of the ablation study in ACM benchmark.

However, beyond difficulty regularization by regret, GEMS has several useful features for multi-agent training. One such is the GenOpt Agent, which provides non-trivial open-loop agents without expert supervision. Compared to the No-OP approach, which involves agents taking no actions, the ablation study shows that training with GenOpt is better. In the ACM environment, for instance, an uncontrolled aircraft's inevitable energy loss leads to a crash. The GenOpt Agent, on the other hand, addresses this issue by providing a sophisticated sequence of actions optimized by the curriculum generator, thereby ensuring sustained engagement and challenge.

Finally, NoVic underscores the critical role of Nash equilibrium in our zero-sum game formulation between the curriculum generator and the ego agent. Unlike NoRegret, NoVic exhibits inferior performance, illustrating that while regret contributes to curriculum generation, $G(\pi_{ego}, \psi)$ is pivotal for ensuring training stability and effective curriculum optimization.

## 6 Conclusion

This paper proposes using a genetic algorithm to improve and stabilize learning in multi-agent environments. Using the strength of genetic algorithms to generate a curriculum in a large scenario space, we enable an RL agent to reach a better solution faster. We also introduce GenOpt Agent, an open-loop agent optimized by the curriculum generator without expert supervision. This enhances training stability, which is especially valuable early on when the self-play opponent lacks sufficient competence. Empirical results across multiple benchmarks show our method outperforming various baselines, and an ablation study further confirms the significance of our design decisions.

## 7 Limitations and Future Works

Similar to MAESTRO [6], our method currently lacks a provable guarantee for convergence to the Nash equilibrium in games. This presents an opportunity to identify conditions under which guaranteed convergence can be achieved.

**Acknowledgments**

This work was partly funded by The Boeing Company.

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

|  | Pong | Volley | ACM |
|---|---|---|---|
| **FSP** | $0.197 \pm 0.158$ | $-0.143 \pm 0.061$ | $94.498 \pm 39.686$ |
| **PSRO** | $0.141 \pm 0.154$ | $-0.145 \pm 0.058$ | $83.274 \pm 51.499$ |
| **GC+FSP** | $0.159 \pm 0.179$ | $-0.004 \pm 0.07$ | $-124.304 \pm 44.356$ |
| **SPDL+FSP** | $-0.386 \pm 0.144$ | $\mathbf{0.192 \pm 0.058}$ | $-97.987 \pm 51.969$ |
| **MAESTRO** | $-0.578 \pm 0.142$ | $-0.094 \pm 0.053$ | $-226.845 \pm 25.332$ |
| **GEMS** | $\mathbf{0.466 \pm 0.121}$ | $\mathbf{0.194 \pm 0.058}$ | $\mathbf{139.080 \pm 32.768}$ |

Table 3: Mean return of algorithms against baseline algorithms and ours for Pong, Volley, and ACM. The highest mean return in bold.

[51] X. Feng, O. Slumbers, Z. Wan, B. Liu, S. McAleer, Y. Wen, J. Wang, and Y. Yang. Neural auto-curricula in two-player zero-sum games. *Advances in Neural Information Processing Systems*, 34:3504–3517, 2021.

[52] T. Haarnoja, A. Zhou, K. Hartikainen, G. Tucker, S. Ha, J. Tan, V. Kumar, H. Zhu, A. Gupta, P. Abbeel, et al. Soft actor-critic algorithms and applications. 2018.

[53] createmind. Deep reinforcement learning. `https://github.com/createamind/DRL/tree/master/spinup/envs/BipedalWalkerHardcore`, 2019.

[54] J. Schulman, P. Moritz, S. Levine, M. Jordan, and P. Abbeel. High-dimensional continuous control using generalized advantage estimation. *arXiv preprint arXiv:1506.02438*, 2015.

[55] N. Barhate. Minimal pytorch implementation of proximal policy optimization. `https://github.com/nikhilbarhate99/PPO-PyTorch`, 2021.

[56] N. Minorsky. Directional stability of automatically steered bodies. *Journal of the American Society for Naval Engineers*, 34(2):280–309, 1922.

# A  Full Performance Results of the Trained Algorithms

In this section, we include the full results of our experiments. Table 3 and Table 4 shows the mean return of each algorithm against each other, while Table 5 and Table 6 includes the detailed results of each pair. Figure 6 shows the training curve, evaluated in round robin fashion.

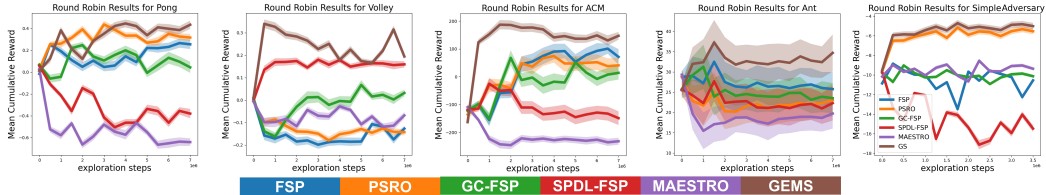

Figure 6: Training curve with round robin results.

# B  Environment Details

This section will discuss the details of the benchmark environments used in this paper.

## B.1  Pong

**Environment Overview**

|         | Ant               | SimpleAdversary    |
|---------|-------------------|--------------------|
| **FSP** | 25.788 ± 4.228    | -10.545 ± 0.166    |
| **PSRO** | 23.061 ± 4.066   | -5.535 ± 0.308     |
| **GC+FSP** | 23.544 ± 3.735 | -10.125 ± 0.158    |
| **SPDL+FSP** | 22.312 ± 4.275 | -15.47 ± 0.338   |
| **MAESTRO** | 19.748 ± 3.706 | -9.358 ± 0.169    |
| **GEMS** | **34.714 ± 4.549** | **-5.010 ± 0.333** |

Table 4: Mean return of algorithms against baseline algorithms and ours for Ant and SimpleAdversary. The highest mean return in bold.

Pong is a game similar to the one described in [42]. In this game, there are two paddles, each located at the left and right edges of the map. Each agent controls a paddle capable of 1D movement. At the start of the game, a ball spawns at the center of the map with a certain velocity. The ball will bounce if it comes in contact with the top edge, the bottom edge, or paddles. An agent wins the game if the ball passes through the opposite edge of the map.

To play the game, each agent observes the position and velocity of the ball and the location of each paddle. The action space is 1-D. If the value is positive, the agent's paddle will move up at a constant speed, and vice versa if negative. An agent receives a reward of +1 if they win and -1 if they lose. If neither player manages to win the game after 300 steps, it is considered a tie with a reward of 0.

### Environment Encoding

The original environment calls the random number generator four times to reset a game. Subsequently, we encode the environment with four values. One value determines whether the ego agent plays the left or right paddle. The other three control the initial velocity of the ball. One value determines whether the ball will be moving up or down, another controls the magnitude of the ball's velocity vector in the up-down axis, and the last one determines whether the ball will be traveling left or right. The magnitude of the ball's velocity in the left-right axis remains constant throughout the game.

### B.2   Volley

### Environment Overview

Volley is an environment based on the concept presented in [43], where two agents engage in a 2D volleyball game. This presents a more intricate update than Pong, as both players can move in a 2D space instead of the 1D movement of paddles. Additionally, the ball's horizontal speed can vary along with its vertical speed. At the start of the game, the ball spawns at the center of the map above the net, which divides the map into two halves. The ball will bounce upon contact with any players or edges except for the bottom. Each agent can redirect the ball by hitting it with the avatar they control. The ball follows a simple physics model, subject to gravity and simple elastic collision mechanics. An agent wins the game by successfully landing the ball on the other side of the map.

To play the game, each agent observes the position and velocity of the ball, the ego agent, and the other agent. Each agent has a 2-D continuous action space, dictating the desired vertical and horizontal velocity of the avatar in the game. While the avatar can move at the desired velocity in the horizontal axis, it can only be launched upward with a desired velocity if it is on the ground. Otherwise, it will move along the surface or be in free fall due to gravity. An agent receives a reward of +1 if they win and -1 if they lose. If neither player manages to win the game after 300 steps, it is considered a tie, and both players receive a reward of 0.

### Environment Encoding

The Volley environment is expressed in three values. The first two values are continuous (-1,1) and define the ball's initial velocity. The third value is discrete 0,1 and defines whether the ego agent controls the avatar on the left or the right.

Table 5: Detailed Win:Tie:Lose Ratio(%) of Each Algorithm Against Others.

**Pong**

| Agent | vs FSP | vs PSRO | vs GC+FSP | vs SPDL+FSP | vs MAESTRO | vs GEMS (Ours) |
|---|---|---|---|---|---|---|
| FSP | 49±1:2±0:49±1 | | | | | |
| PSRO | 42±1:1±0:57±1 | 49±1:3±0:49±0 | | | | |
| GC+FSP | 48±1:1±0:51±1 | 50±1:0±0:49±1 | 50±1:0±0:50±1 | | | |
| SPDL+FSP | 22±0:0±0:78±0 | 22±1:1±0:78±1 | 19±0:0±0:80±0 | 50±0:0±0:50±0 | | |
| MAESTRO | 9±0:0±0:91±0 | 13±1:0±0:87±1 | 40±0:0±0:60±0 | 50±0:0±0:50±1 | 50±0:0±0:50±0 | |
| GEMS (Ours) | 66±1:4±0:30±0 | 63±1:4±0:33±1 | 73±1:2±0:25±1 | 89±0:1±0:11±0 | 94±0:0±0:6±0 | 47±1:7±0:47±1 |

**Volley**

| Agent | vs FSP | vs PSRO | vs GC+FSP | vs SPDL+FSP | vs MAESTRO | vs GEMS (Ours) |
|---|---|---|---|---|---|---|
| FSP | 50±0:0±0:50±0 | | | | | |
| PSRO | 50±0:0±0:50±0 | 50±0:0±0:50±0 | | | | |
| GC+FSP | 58±0:0±0:42±0 | 59±0:0±0:41±0 | 49±0:2±0:49±0 | | | |
| SPDL+FSP | 59±0:16±0:25±0 | 59±0:15±0:26±0 | 51±0:21±0:29±0 | 30±0:41±0:30±0 | | |
| MAESTRO | 51±0:0±0:49±0 | 52±0:0±0:48±0 | 46±0:0±0:54±0 | 29±0:15±0:56±0 | 50±0:0±0:50±0 | |
| GEMS (Ours) | 61±0:13±0:26±0 | 58±0:17±0:25±0 | 54±0:20±0:27±0 | 31±0:37±0:31±0 | 54±0:20±0:27±0 | 31±0:38±0:31±0 |

**ACM**

| Agent | vs FSP | vs PSRO | vs GC+FSP | vs SPDL+FSP | vs MAESTRO | vs GEMS (Ours) |
|---|---|---|---|---|---|---|
| FSP | 25±0:49±0:25±0 | | | | | |
| PSRO | 20±0:54±0:26±0 | 22±0:55±0:22±0 | | | | |
| GC+FSP | 3±0:48±0:49±0 | 3±0:50±0:47±1 | 38±2:23±1:38±2 | | | |
| SPDL+FSP | 2±0:48±0:50±0 | 2±0:49±0:49±0 | 63±2:23±1:15±1 | 8±0:84±0:8±0 | | |
| MAESTRO | 1±0:39±0:60±0 | 10±1:13±0:78±1 | | 2±0:5±0:93±0 | 47±1:6±0:47±1 | |
| GEMS (Ours) | 32±0:56±0:14±0 | 50±0:48±0:2±0 | 48±0:50±0:2±0 | 48±0:50±0:1±0 | 62±0:37±0:1±0 | 18±0:64±1:18±0 |

**Ant**

| Agent | vs FSP | vs PSRO | vs GC+FSP | vs SPDL+FSP | vs MAESTRO | vs GEMS (Ours) |
|---|---|---|---|---|---|---|
| FSP | 36±0:27±0:37±0 | | | | | |
| PSRO | 34±0:23±0:43±0 | 43±0:15±0:42±0 | | | | |
| GC+FSP | 31±1:27±0:42±1 | 40±1:20±1:40±1 | 42±0:20±0:38±0 | | | |
| SPDL+FSP | 32±0:22±0:45±0 | 48±0:17±0:41±0 | 38±1:20±0:42±1 | 43±0:16±0:41±0 | | |
| MAESTRO | 30±0:25±1:45±0 | 36±1:23±1:41±1 | | 39±0:18±0:42±0 | 42±1:18±0:41±1 | |
| GEMS (Ours) | 36±0:42±1:22±1 | 47±1:31±1:22±1 | 51±0:30±1:18±1 | 49±1:31±1:20±1 | 51±0:30±1:18±1 | 26±1:47±1:27±1 |

Table 6: Detailed Win:Tie:Lose Ratio(%) of Each Algorithm Against Others. (SimpleAdversary)

| Agent | vs FSP | vs PSRO | vs GC+FSP | vs SPDL+FSP | vs MAESTRO | vs GEMS (Ours) |
|---|---|---|---|---|---|---|
| | | | SimpleAdversary | | | |
| FSP | 50±0 :0±0 :50±0 | | | | | |
| PSRO | 53±0 :0±0 :47±0 | 50±0 :50±0 :50±0 | | | | |
| GC+FSP | 50±0 :0±0 :50±0 | 46±0 :0±0 :54±0 | 50±0 :0±0 :50±0 | | | |
| SPDL+FSP | 46±0 :0±0 :54±0 | 43±0 :0±0 :57±0 | 47±0 :0±0 :53±0 | 50±0 :0±0 :50±0 | | |
| MAESTRO | 49±0 :0±0 :51±0 | 47±0 :0±0 :53±0 | 49±0 :0±0 :51±0 | 52±0 :0±0 :48±0 | 50±0 :0±0 :50±0 | |
| GEMS (Ours) | 54±0 :0±0 :46±0 | 50±0 :0±0 :50±0 | 54±0 :0±0 :46±0 | 57±0 :0±0 :43±0 | 54±0 :0±0 :46±0 | 53±0 :0±0 :47±0 |

## B.3 ACM

### Environment Overview

ACM is a simulated dogfight environment. This is more complex than the previous two benchmarks, as ACM agents will move in 3D space with realistic physics simulation. At the start of the game, two aircraft spawn in the air at a certain orientation and a velocity vector with respect to the nose heading. The agent wins the game by pointing its nose at the opponent while avoiding getting pointed by the opponent's nose. There is also a penalty involved with crashing to the ground.

To play the game, each agent observes the position, orientation, and velocity of the ego and the opponent agents. Each agent has a 4-D continuous action space with direct control over the aircraft's elevator, aileron, rudder, and throttle. The physics is simulated by JSBSim [44], a high-fidelity simulator often used in autonomous aircraft research. The aerodynamics model for ACM is based on a Boeing F-15D, capable of flying two times faster than the speed of sound.

To win the game, the agent should have its nose pointed less than 5 degrees off the opponent aircraft while flying less than 2,000 feet away from the opponent. The agent can also win the game if the opponent aircraft flies below the hard deck of 500 ft. The agent wins the game if either of these conditions are met and loses if the opponent achieves either of the conditions. It is a tie if both aircraft achieve either of these conditions simultaneously. If either of the agents meets none of the conditions for 300 seconds, it is considered a tie. Agent receives a penalty of -300 for crashing into the ground, 150 for pointing the nose toward the target, and 100 if the opponent crashes to the ground. To help with training, there is a small, dense reward for getting closer to pointing the nose toward the opponent.

### Environment Encoding

The ACM environment is expressed in 14 continuous values. They define the initial conditions of the game, which are each aircraft's position, orientation, and airspeed.

## B.4 Ant

### Environment Overview

Ant is a simulated Mujoco environment where two ant agents try to flip or push the other agent out of the arena. With each agent having 8 degrees of freedom in a contact-rich setup, the environment offers a variety of behaviors to learn. For example, while ramming is often the only way to win against a stationary target, ramming is vulnerable to flipping attacks. However, flipping attacks are useless against stationary opponents as agents need more torque to lift and flip an opponent with all its legs firmly contacting the ground.

To play the game, each agent observes the position, orientation, velocity, and posture of the ego and the opponent agents. Each agent has 8-D continuous action space with direct control over the torques applied to each joint. The agents get a reward of 100 if the opponent is flipped or pushed out of the arena. There is a penalty of -20 if the agent is flipped or exits the arena. To make the training faster, there is a sparse reward for approaching the opponent or moving towards the center of the arena. If neither player manages to win the game after 300 steps, it is considered a tie, and both players receive a reward of 0. While the environment is based on OpenAI's implementation of Sumo-Ant [11], the arena is smaller to allow agents to interact more and learn faster.

### Environment Encoding

The Ant environment is expressed in 1 discrete value assigning who will be controlling the red agent and who will be controlling the blue agent in the game. The red and blue agents have the same winning and losing conditions and have fixed spawn points in the arena.

### B.5 SimpleAdversary

SimpleAdversary is a Multi Particle Environment from [46]. SimpleAdversary provides an interesting opportunity to validate our GEMS on environments that are asymmetrical and have both competitive and collaborative features to a game. **??** shows a screenshot of the environment.

In this environment, there is one adversary agent (red), two good agents (blue), and two landmarks (green and black). One of the landmarks is a target landmark where the good agents get a reward for the good agents getting close to the agent and a negative reward for the adversary agent getting close to the target landmark. The vice versa applies to the reward for the adversary agent. While all agents can see the location of each other and landmarks, only good agents have additional information on where the goal is. The good agents observe the environment with 10-D space including the goal information while the adversary agent observes the environment with 8-D space missing this environment. The action space is continuous, with the agents being allowed to move up, down, left, and right.

This environment covers challenges that are not covered well in the original manuscript. First, it is an asymmetrical environment where the goal and the optimal policies differ by role. Also, there are two good agents in the game which has to learn to collaborate with each other. Finally, there are three agents in this game, giving a limited yet first glimpse into scalability regarding the number of agents.

### B.6 Environment Encoding

SimpleAdversary has a total of 5 objects in the game consisting of two landmarks, two good agents, and one adversary agent. The first 10 variables in the environment encoding cover the X and Y coordinates of the objects in the game. The last 11th variable in the environment encodes whether the ego agent is playing as the good agent or the adversary agent in the game.

As observation space is different whether playing as an adversary agent or a good agent, we added a zero padding to equalize the dimensions. In addition, we also add a variable to the observation space denoting whether the agent is playing as an adversary agent or a good agent. This allows the policy network to run as a conditional network able to support both roles in the game.

## C  Implementation Details

This section covers the details of implementing the training procedures for each baseline for reproducibility purposes.

### C.1  Training Hardware

We used an 88-core Intel Xeon Gold 6238 CPU at 2.10 GHz to train the models. Training models took from $5 \sim 15$ days. Lighter benchmarks, like Pong, took 5 days, while heavier benchmarks, such as ACM, took 15 days. This is due to the computation load of the JSBsim, which provides high-fidelity aerodynamics simulation.

### C.2  Tuning RL Algorithm

**Tuning Procedures**

To ensure that hyperparameters do not favor one baseline over another, they were tuned for the RL policy explorer in a single-agent setup against hand-coded opponents. For Pong and Volley, the opponents were PID controllers chasing the estimated impact point of the ball [56]. In the case of ACM, a simple PID autopilot was employed to control the opponent aircraft's speed, heading, and altitude. Ant was tuned with a stationary agent that does not move. The combination of network structure and hyperparameters that performed the best in these single-agent setups was selected for our baseline experiments. Note that these hand-coded agents were only used to tune the RL

| Benchmark | Parameter Name | Range |
|---|---|---|
| **Pong** | Magnitude of the Ball's Vertical Speed | [0,1] |
| | Left-Right Initial Velocity Direction of the Ball | {0,1} |
| | Up-Down Initial Velocity Direction of the Ball | {0,1} |
| | Left / Right Paddle Controlled by the Ego Agent | {0,1} |
| **Volley** | Initial Horizontal Velocity of the Ball | [0,1] |
| | Initial Vertical Velocity of the Ball | [0,1] |
| | Left / Right Paddle Controlled by the Ego Agent | {0,1} |
| **ACM** | Ego Agent Initial X Coordinates | [-1,1] |
| | Ego Agent Initial Y Coordinates | [-1,1] |
| | Ego Agent Initial Z Coordinates | [-1,1] |
| | Ego Agent Initial Heading | [-1,1] |
| | Ego Agent Initial Pitch Angle | [-1,1] |
| | Ego Agent Initial Role Angle | [-1,1] |
| | Ego Agent Initial Airspeed | [-1,1] |
| | Opponent Agent Initial X Coordinates | [-1,1] |
| | Opponent Agent Initial Y Coordinates | [-1,1] |
| | Opponent Agent Initial Z Coordinates | [-1,1] |
| | Opponent Agent Initial Heading | [-1,1] |
| | Opponent Agent Initial Pitch Angle | [-1,1] |
| | Opponent Agent Initial Role Angle | [-1,1] |
| | Opponent Agent Initial Airspeed | [-1,1] |
| **Ant** | Ego and Opponent Agent Spawnpoint Configurations | {0,1} |
| **SimpleAdversary** | Good Agent 0 Location, X | [-1,1] |
| | Good Agent 0 Location, Y | [-1,1] |
| | Good Agent 1 Location, X | [-1,1] |
| | Good Agent 1 Location, Y | [-1,1] |
| | Adversary Agent 1 Location, X | [-1,1] |
| | Adversary Agent 1 Location, Y | [-1,1] |
| | Goal Location, X | [-1,1] |
| | Goal Location, Y | [-1,1] |
| | Obstacle Location, X | [-1,1] |
| | Obstacle Location, Y | [-1,1] |
| | Ego Agent is Playing As | {0,1} |

Table 7: Summary of environmental parameters.

hyperparameters. The hand-coded agents were not used when training the baselines and our approach, GEMS.

### Network Structure

To train our RL algorithm, we experimented with various network structures, varying the depth of the hidden layers from 1 to 4 layers and the width from 64 to 512. Our findings revealed that the optimal architecture for Pong, Volley, and ACM consists of hidden layers composed of two fully connected layers, each with a size of 256. We utilized ReLU activation between the hidden layers and concluded that the last layer had an output dimension equivalent to the action dimension, followed by Tanh activation. Conversely, Ant uses 2-layer policy networks with Tanh activation and a width of 128.

### C.3 RL Hyperparameters

We conducted a limited grid search to hyperparameters for our RL algorithm. For learning rate, we tried {3e-5, 1e-4, 3e-4, 1e-3}. For discount ratio $\gamma$, we tried {0.95, 0.98, 0.999}.

| | Pong | Volley | ACM | Ant | SimpleAdversary |
|---|---|---|---|---|---|
| **Start Steps** | 10000 | 10000 | 10000 | N/A | 10000 |
| **Learning Rate** | 3e-4 | 3e-4 | 3e-4 | 3e-4 | 3e-4 |
| $\gamma$ | 0.98 | 0.98 | 0.98 | 0.98 | 0.98 |
| $\alpha$ | auto | auto | auto | N/A | auto |
| **Batch Size** | 256 | 256 | 512 | 36864 | 256 |
| **Replay Size** | 1e6 | 1e6 | 1e6 | N/A | 1e6 |
| **Update Every** | 1 | 1 | 1 | 36864 | 1 |
| **Backpropagation per Update** | N/A | N/A | N/A | 20 | N/A |

Table 8: Selected hyperparameters for RL algorithms.

Regarding the algorithms that use SAC, we tried 0, 1000, 10000 for the initial exploration steps. For batch size, we tried {64, 256, 512}. For update every, we tried {1, 32, 100}. We tried {1e6, 3e6} for Replay size. We ran each setting on 3 seeds to balance performance and computational cost.

Regarding algorithms that use PPO, we tried policy update frequency of 9216,18432,36864. For the number of times the gradient was calculated and backpropagated per update frequency, we tried 10,20,30. We ran each setting on 3 seeds to balance performance and computational cost.

Table 8 shows the selected hyperparameters we used for the main experiments.

## D   Baseline Hyperparameter Tuning

To tune the hyperparameters for the baselines, we trained the models using the following settings. Since it is challenging to tune the hyperparameters by conducting a round-robin across all hyperparameter settings used for different baselines, each setting was evaluated against the same hand-crafted opponents used for tuning the RL hyperparameters. Striking a balance between performance and computational cost, we ran each configuration with 3 seeds. Table 9 presents the selected hyperparameters used for the main experiments.

We tested the following hyperparameter values in a limited grid search. For adding checkpoints, we tried saving a copy of the agents' policies at every {10000, 30000, 50000} exploration step. We used the same interval of steps as the interval in which curriculum is generated for the algorithms that generate one at every epoch. For measuring performance between each checkpoint on random environment parameters to approximate the Nash equilibrium in PSRO, we tried {10,30} evaluations per pair. As for the hyperparameters of the GC, we tried {200,300} for the size of the population of scenarios evaluated for whether being solved or not solved. We also tried sizes of {200,300} for the sizes of the generated curriculum and mutation rates of {0.1,1}. For SPDL, we tried values of {0,10,20} for offset and {0.1,0.3,1} for penalty proportion. For MAESTRO, we tried co-player exploration coefficients of {0.05,0.1} and curriculum buffer sizes of {500,1000}. For GEMS, we copied the hyperparameters from the GC in terms of the size of the curriculum and mutation rate. This was done to make comparisons between different algorithms easier by removing the effect of different hyperparameters during genetic operations.

## E   Crossover and Mutation

This section covers how GEMS perform crossover and mutation. Please note that these steps simply conduct generic genetic operations and are not hand-tuned nor hardcoded to specifically solve the benchmark tasks in this paper.

For a sampled parent scenarios $\mathbf{m}, \mathbf{n}$, the corresponding encoding would look like;

$$\mathbf{m} = \{i_\pi, \pi_\varnothing, \psi\} = \{m_0, m_1\} = \{m_{0,0}, m_{0,1}...m_{0,x}, m_{1,0}, m_{1,1}...m_{1,y}\}$$

|  | Pong | Volley | ACM | Ant |
|---|---|---|---|---|
| **PSF** | | | | |
| **Checkpoint Interval** | 10000 | 10000 | 10000 | 50000 |
| **PSRO** | | | | |
| **Match Times** | 30 | 30 | 10 | 10 |
| **GC** | | | | |
| **Evaluation Set Size** | 200 | 200 | 300 | 300 |
| **Curriculum Size** | 200 | 200 | 300 | 300 |
| **SPDL** | | | | |
| **Penalty Proportion** | 1 | 1 | 0.1 | 0.1 |
| **Offset** | 20 | 20 | 20 | 20 |
| **MAESTRO** | | | | |
| **Co-Player Exploration Coefficient** | 0.1 | 0.1 | 0.1 | 0.1 |
| **Curriculum Buffer Size** | 1000 | 1000 | 1000 | 1000 |

Table 9: Selected hyperparameters for baselines.

$$\mathbf{n} = \{i_\pi, \pi_\varnothing, \psi\} = \{n_0, n_1\} = \{n_{0,0}, n_{0,1}...n_{0,x}, n_{1,0}, n_{1,1}...n_{1,z}\}$$

$\mathbf{m}_o$ encodes the choice of opponent to load $i_\pi$ and the environment parameters $\psi$ while $\mathbf{m}_o$ represents the encoding of the GenOpt Agent $\pi_\varnothing$ .In this section, $x$ corresponds to the length of environment encoding for $\mathbf{m}$. $y$ and $z$ in this section corresponds to the length of GenOpt Agent $\pi_\varnothing$ encoding for $\mathbf{m}$ and $\mathbf{n}$. If using the same environment, $x$ will not be different between the scenarios, whereas $y$ and $z$ will vary depending on the encoded environment.

GEMS performs crossover by swamping a section from one parent with a section from another parent. This would be done by randomly selecting three split points in the encoding. Using a uniform distribution, GEMS sample three integer values $\eta_0 \in \{0, 1, 2, ..x\}, \eta_1 \in \{0, 1, 2, ..y\}, \eta_2 \in \{0, 1, 2, ..z\}$

From this, GEMS generates two child scenarios $\mathbf{p}, \mathbf{q}$ as follows;

$$\mathbf{p} = \{m_{0,0}...m_{0,\eta_0}, n_{0,\eta_0+1}...n_{0,x}, m_{1,0}...m_{1,eta_1}, n_{1,eta_2+1}...n_{1,z}\}$$

$$\mathbf{q} = \{n_{0,0}...n_{0,\eta_0}, m_{0,\eta_0+1}...m_{0,x}, n_{1,0}...n_{1,eta_2}, m_{1,eta_1+1}...n_{1,y}\}$$

For the special cases such as $\eta_0 = 0, x$ or $\eta_1 = 0, y$, or $\eta_2 = 0, z$, GEMS inherits the segment without dividing it. For example, if $\eta_0 = 0$, then $\mathbf{p_0} = \mathbf{n_0}, \mathbf{q_0} = \mathbf{m_0}$ and so on.

To perform a mutation on a sampled scenario, GEMS first performs a crossover between the scenario and a randomly generated scenario and returns one of the child scenarios as the mutated scenario. The mutation rate controls the probability of each scenario undergoing mutation.

## F   Sampling Parents

In this paper, we have defined the probability of GEMS sampling scenarios as parents as the following;

$$p(\xi) \propto \delta(\xi)(1 - G(\pi_{ego}, \xi))$$

In practice, GEMS uses the normalized form as follows;

$$p(\xi_i) = \frac{\delta(\xi_i)(1 - G(\pi_{ego}, \xi_i))}{\Sigma\delta(\xi)(1 - G(\pi_{ego}, \xi))}$$

If $\sum \delta(\xi) = 0$, GEMS use regret-only version as follows;

$$p(\xi_i) = \frac{(1 - G(\pi_{ego}, \xi_i))}{\Sigma(1 - G(\pi_{ego}, \xi))}$$

## G   Evolution of Curriculum

Figure 7 includes characteristic examples of the scenarios generated and used as curriculum for each algorithm in the ACM benchmark.

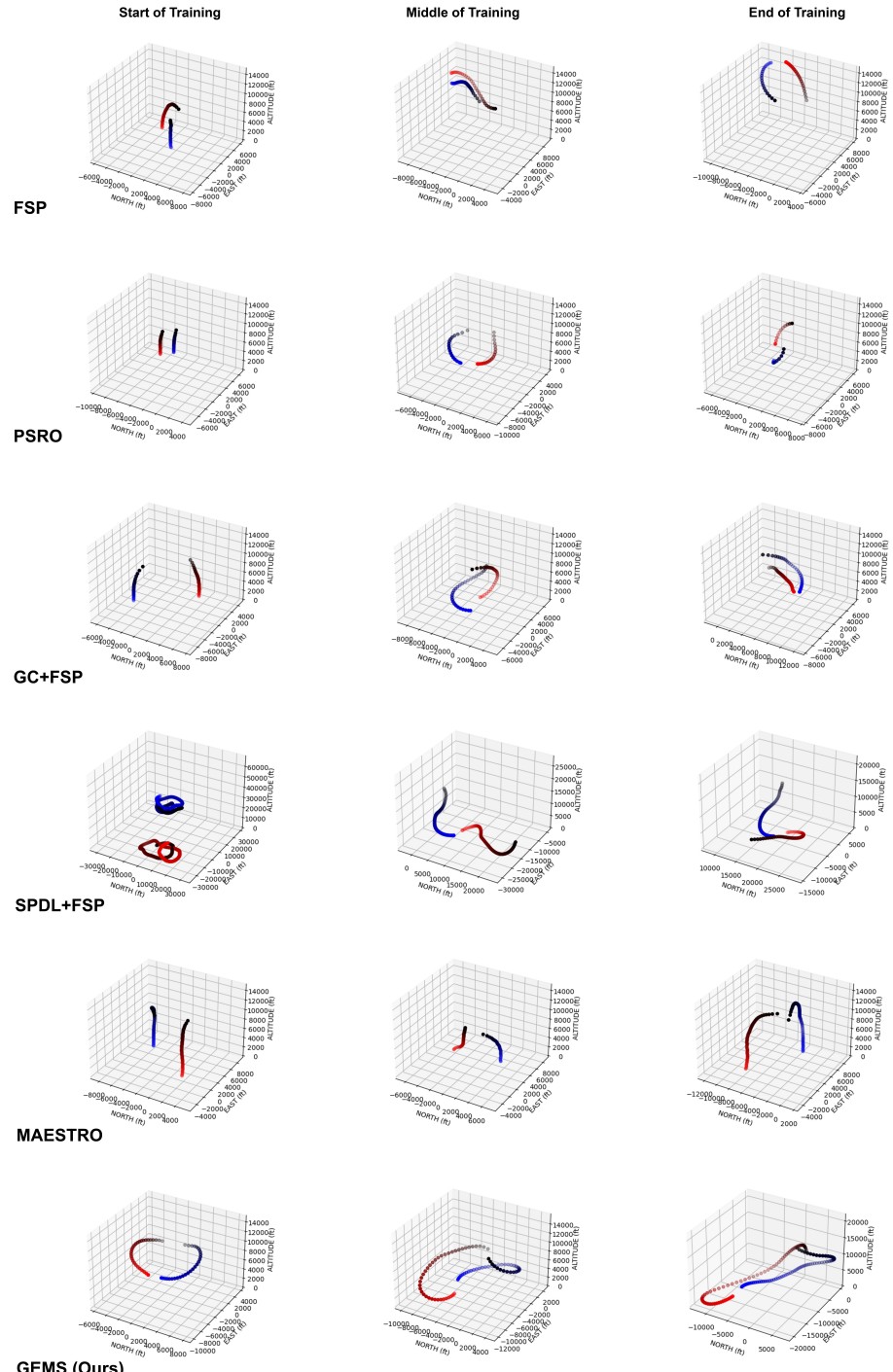

Figure 7: Each dot marks the position of the red and blue aircraft at 1-second intervals. The color of the markers transitions from black to blue for the student aircraft and from black to red for the opponent aircraft. It is important to note that at the start of the training, only our approach, GEMS, can present scenarios and opponents that provide interesting data points instead of simply crashing to the ground. While SPDL+FSP somewhat succeeds in finding scenarios that do not end in a crash, it optimizes scenarios without considering opponent policy. It simply finds trivial cases where agents are flying in circles.

