# OpenReview forum: "Genetic Algorithm for Curriculum Design in Multi-Agent Reinforcement Learning"
_robot-learning.org/CoRL/2024/Conference — CoRL 2024_

### Official Review · Reviewer_jtXR · 2024-07-11
**Review of Genetic Algorithm for Curriculum Design in Multi-Agent Reinforcement Learning**

**Originality:** 3
**Technical Quality:** 4
**Clarity Of Presentation:** 5
**Potential Impact:** 2
**Recommendation:** 4
**Confidence:** 4

**Review:**

**Quality and Clarity**

The quality of the writing is high and the paper was for the most parts easy to read. The presentation is clear and concise with well done figures to help explain the underlying concepts. There are some questions remaining regarding the approach and the evaluation.

** Originality **

The paper introduces a new framework using genetic algorithms for curriculum generation. While the concept of curriculum generation is known, the combination of genetic modification and regulation of difficulty of scenarios is novel making this work an incremental improvement over existing work.

** Significance **

The significance of this paper is limited. This is because no theoretical guarantees are made and the range of scenarios this approach was shown on is limited to competitive multi-agent environments. Nevertheless, this work sets a new baseline in this niche.

** Strenghts **

- Clarity of idea
- Empirical Validation
- Extensive ablation studies

** Weaknesses **

- Limited scenario range (only competitive)
- Unclear Evaluation regarding negative aspects of the approach

**Quality Of The Limitations Section:**

2

**Questions For Rebuttal:**

Question 1:
In the evaluation section you write that you evaluate in a round robin format of the agents. However as GEMS still loses to others it does provable never learn an optimal strategy to cover all opponent strategies. What are the reasons for the exploitable weaknesses that remain in the learning process?

Question 2:
If I understood correctly, the regret component of GEMS chooses environments, which have the regret of -1 with a higher probability to reuse (given modifications). How is it ensured that environments of appropriate difficulty are selected? Could not the hardest difficulties, which yield a high regret but also are too hard for learning currently and up being selected making learning very expensive? What if the space of learnable scenarios is small in comparison to those which are too hard to learn?

**Robotics Focus:**

1

**Summary Of Paper:**

The paper introduces a genetic algorithm approach to optimize the curriculum of learning agents in simulated environments with multiple agents using self-play with checkpoints. Their approach utilizes a scenario generator which is able to perform traditional genetic algorithm operations given some scenario and the regret of the policy in past scenarios. The authors formulate the learning policy and this generator as adversaries in a zero sum game. Basically, the generator tries to find the scenarios and modifies them through mutation, in which the current policy performs bad and the current policy tries to maximize its return on the current set of given scenarios. Opposing players in each scenario are sampled by the generator from past checkpoints of the policy. This approach is tessted in four environments and evaluated against the PSRO baseline. The new approach GEMS performs best in all environments and shows robustness in evaluation against 5 baseline agents and the GEMS agent. They also provide an ablation study showing that the regret component of their algorithm is not important for early training but maximizing out an agents final performance.

**Summary Of Recommendation:**

The paper is an enjoyable read and presents clearly a new idea, that is a combination of previously known concepts applied in an interesting manner. The approach outperforms existing methods and therefor is of interest to the scientific community. While there are minor improvements that can be made to the paper i recommend accepting.

---

### Official Review · Reviewer_FqYm · 2024-07-20
**Review of Submission 545**

**Originality:** 3
**Technical Quality:** 4
**Clarity Of Presentation:** 3
**Potential Impact:** 3
**Recommendation:** 3
**Confidence:** 5

**Review:**

## Strength:
1. The paper is well-written and introduces most of the key concepts in the proposed methodology clearly.
2. The experiments are extensive and show positive empirical results of the proposed approach.

## Weakness:
1. A big limiting factor of the proposed approach is it’s only designed for symmetric agents in symmetric environments, namely the ego and opponent agents have the same observation and action space. The baselines being compared are more general in this aspect.
2. The problem that the paper is trying to solve is somewhat nebulous, the proposed approach, and the problem formulation itself is aimed towards curriculum learning where environment the agent is in is part of the curricular, while the baseline being compared (PSRO) is aimed towards finding approximated Nash strategy profiles for both the ego and the opponent, while less focused on the environment.
3. The proposed methodology has limited novelty since crossover/mutation is well studied in prior genetic algorithm research, and the estimated regret was introduced by prior work. Moreover, the study is mainly empirical, while a big part of the problem the paper is trying to tackle is computational efficiency when compared to other baselines, and convergence/sample complexity analysis is important to support the claims.

## Additional Comments:
1. Since the regret is estimated, and highly biased towards the current ego policy, and from the ablation results, the argument for the necessity is weak. The plateauing in Figure 5 doesn’t seem to be that important for the overall performance. It might be helpful to show the regret curve as well.
2. While I understand the need for the GenOpt agent, the implementation is not clearly described. See Question 2 for more details.

## Minor Comments:
1. In Appendix E, the GenOpt agent is referred to as the Blind Agent.
2. When introducing the estimated regret, it’ll be better to add an intuitive description of what it’s measuring (gap from the current obtained return to the maximal possible return of the level).

**Quality Of The Limitations Section:**

3

**Questions For Rebuttal:**

1. I would love to hear some rebuttal to points 1 and 2 that I’ve written down in the weakness section.
2. How are the GenOpt agents parameterized? Does GEMS directly search over the actions at each timestep range? Are the timestep range knotpoints (t1, t2, etc.) chosen beforehand and fixed? Since the parameterization of the GenOpt agent is always present as part of the genome, wouldn’t the outcome obtained when NOT using the GenOpt agent introduce potentially wrong direction for optimization in GEMS?

**Robotics Focus:**

2

**Summary Of Paper:**

This paper introduces a curriculum generation scheme for multiagent scenarios using genetic algorithms. The authors show positive results through extensive experiments.

**Summary Of Recommendation:**

Although showing extensive and positive experimental results, I recommend a weak rejection due to concerns on limited novelty in the proposed methodology, no theoretical analysis on convergence and sample complexity, and limited generalization to asymmetrical problems. After rebuttal, the authors sufficiently addressed the biggest concern I had during review, hence I've updated my recommendation to weak accept.

---

### Official Review · Reviewer_7Pcd · 2024-07-30

**Originality:** 3
**Technical Quality:** 3
**Clarity Of Presentation:** 4
**Potential Impact:** 3
**Recommendation:** 3
**Confidence:** 4

**Review:**

The method as described seems relatively simple, and the results show an improvement over the baselines. I particularly like the justification of experiment and baseline selection, and attempts at fairness in the results.

I found this paper was well written, though there were critical details of the method that weren’t clear to me (questions below). Overall I’m not convinced  of the novelty of the contribution, though this might be because I haven’t understood some of the key elements. There doesn’t appear to be a limitations section, I’m not sure if that is still a requirement for this conference. It’s also not a requirement to have real robot experiments though sim only experiments are harder to show novelty.

The method states the GenOpt agent is an optimised open-loop trajectory using genetic algorithms without supervision, but I couldn’t see any more details (even in the appendix) about the training of these. What are the specifics for each environment, what is the GenOpt Agent script (Fig1)? Similarly, what are some of the environment attributes that are mutated for each experiment? My apologies if I overlooked these.

What could be the cause of the larger number of ties compared to wins/losses? MAESTRO has significantly fewer ties for all tasks than GEMS. Perhaps GEMS formulates safer behaviours that prevent losing but are not as diverse as those generated through random exploration?

Fictitious Self Play seems to be a strong baseline on all tasks. On what tasks does self-play induce instability?

Formatting of equation at line 103 defining the expected return objective should include subscripts for action sampling.

“We train each algorithm for 7e6 steps in each benchmark for each benchmark” -> Double up

“regret term (σ)” (line 281) -> regret term was a delta (δ)?

Update reference list with peer reviewed sources where possible, e.g:

T. Bansal, J. Pachocki, S. Sidor, I. Sutskever, and I. Mordatch. Emergent complexity via multi-agent competition. arXiv preprint arXiv:1710.03748, 2017. -> from ICLR 2018

M. Samvelyan, A. Khan, M. Dennis, M. Jiang, J. Parker-Holder, J. Foerster, R. Raileanu, and T. Rocktäschel. Maestro: Open-ended environment design for multi-agent reinforcement learning. arXiv preprint arXiv:2303.03376, 2023. -> ICLR 2023

**Quality Of The Limitations Section:**

2

**Questions For Rebuttal:**

See above

**Robotics Focus:**

2

**Summary Of Paper:**

This paper introduces GEnetic Multi-agent Self-play (GEMS), a genetic curriculum to select environment variables and opponents, and GenOpt agent, an open-loop opponent that is optimised by the curriculum generator. GEMS outperforms several baselines on multi-agent reinforcement learning tasks (Pong, Volley, ACM, Ant).

**Summary Of Recommendation:**

I think there is some interesting work here, though I'm not convinced the contribution is sufficient. -> Increased score after rebuttal.

---

### Official Review · Reviewer_1f7s · 2024-08-02
**The paper is well written and easy to follow. However, it is not clear if the result is statistically significant.**

**Originality:** 2
**Technical Quality:** 3
**Clarity Of Presentation:** 3
**Potential Impact:** 2
**Recommendation:** 2
**Confidence:** 5

**Review:**

Strength:
- The paper is well written and easy to follow.
- The paper provided an extensive experiments and comparison with the previous work.
- The paper considers Genetic Algorithm for curriculum learning, which is different from most of RL methods and  outperforms previous work in different competitive benchmarks.
Weakness:
- The results shows the proposed method (GEMS) outperforms all the baselines. However, it is not clear if the result are statistically significant.
- It is not clear if the proposed method can be considered for robotics application. All of the experiments are in simulation and it is more in the domain of game environment. They do not include any physical or dynamic features and constraints which limits this work.
- It is not clear how GenOpt can be more robust to adversarial opponents
- More ablation study and experiment for GenOpt is required
- There are some typos and punctuations that needs to be addressed during the rebuttal (please the reviewer's comment in rebuttal)
- It is mentioned the difference between the GC baseline and the proposed method is that GC baseline has a separate evaluation step, while GEMS evaluation is during training., which would reduce the complexity.  However, there is no evidence/result show that how this difference makes GEMS more desirable

**Quality Of The Limitations Section:**

2

**Questions For Rebuttal:**

1- The related work only provides previous work and their limitation. It is not clear how the proposed method fills their gap and limitation, which needs to be provided in related section
2- Figure 1: it is not clear if Ego student is GenOpt agent. Also, why having GenOpt similar to ego agent is beneficial?
3- line 120: 2 player-> 2-player or two player
4- Algorithm 1, line 14: reward -> regret?
5- It would be helpful if the author/s provide a pseudo code or algorithm for GenOpt Agent, or they improve the explanation.
6- Line 103: it seems it provides a formulation for  RL with deterministic assumption as there is not transition function in  this formulation
7- Table 1: it would be helpful if the author/s provide statistical result to show whether their method is statistically significant
8- To add more clarity, the reviewer suggests the word' (ours)' infront of the proposed method
9- It is not clear why those games are selected and how this environments are challenging, in other words, why the proposed method would achieve better result.
10- Figure 5: it seems GEMS and MEASTRO would perform similarly when they converge, the reviewer suggest  running more episode to see their trend when they converge.

**Robotics Focus:**

2

**Summary Of Paper:**

This paper presents a genetic algorithm based method for curriculum lear.ning in multi-agent RL setting. It also introduces a genetically optimized agent, GenOpt Agent, which executes a scheduled action. This agent is optimized without an expert supervision . The result shows the proposed method outperform the previous work in two-player competitive setting.

**Summary Of Recommendation:**

The reviewer suggests major modification to add more clarity in methodology. The reviewer's concern is  the result being statistically significance compared to the previous work.

---

### Author Rebuttal · Authors · 2024-08-09

We sincerely thank the reviewers and the area chair for their insightful questions and feedback! We are encourage that the reviewers found our paper to be well written **(R2, R3)** and easy to follow **(R1)** with clear idea **(R4)**, experiments to be extensive **(R1,R3)** and shows improvement over baselines **(R1, R2,R3)**. showing empirical validation **(R4)**. We are excited that our reviewers liked our justification of experiments and baseline selection and attempts at fairness in results **(R2)** and our ablation study to be extensive **(R4)**.

We also found the reviewer's questions and suggestions raised on stylistic, theory, and related works to be very helpful. Other than stylistic recommendations, notations, clarifications and typos pointed by the reviewers, we also edited the related works to add explanation on how we address the limitations of the previous works, cited a new related works in genetic algorithms for curriculum generation. The attached **PDF** shows major edits in the revised manuscript coded in blue.

Some of our reviewers suggested pseudocode for better explanation. We agree that this would make our paper better and we have included **pseudocode.txt** as a new supplemntary material.

Also, there were points raised that the paper lacks a seperate limitations section. We therefore summarized the limitations we mentioned in the paper, along with points suggested by our reviewers, to formulate a new Limitations and Future Works section. listed as **Section 7** in the paper.

We appreciate the reviewer's suggestions and the rebuttal process to make our paper better! We hope we have been able to address your feedbacks in the revised manuscript, pseudocodes, and our rebuttal responses. Please kindley let us know if you have any more questions or concerns that stands between us and a higher score.

Reviewr IDs : **R1** (1f7s), **R2** (7Pcd), **R3** (FqYm), **R4** (jtXR)

---

### Decision · Program_Chairs · 2024-09-04

**Decision:**

Accept

**Comment:**

The reviewers all saw some strengths with the paper, however the authors must address multiple comments and perform numerous revisions to improve chances of acceptance.

The work is interesting, generally clear, and (to some extent) novel.  Compared to the presented baselines, and with the constraint of only being suitable for symmetric scenarios, the algorithm presents some promising results.

However, it is rather incremental and - in addition to the comments on novelty highlighted by the reviewers, I can point the authors to https://dl.acm.org/doi/abs/10.1145/3512290.3528870 for a related approach.  The constraint of limiting to only symmetric scenarios is significant and only represents a small minority of the types of scenarios we'd like an curriculum design algorihtm like this to work on.

Due to the incremental nature of the work, there are questions around the level of contribution provided by the paper.

The paper also lacks a proper discussion of limitations, and is only done on simulation experiments with no real-world testing.



Update:

The authors have done a good job in responding to reviewer comments, and have effectively answered the main issues with the original paper.  The paper is significantly improved.